# Microbial Complexity of Oral Cavity of Healthy Dogs Identified by Mass Spectrometry and Next-Generation Sequencing

**DOI:** 10.3390/ani13152467

**Published:** 2023-07-31

**Authors:** Fábio V. R. Portilho, Juliano Nóbrega, Beatriz O. de Almeida, André R. Mota, Carolina L. de Paula, Fernando J. P. Listoni, Sandra M. G. Bosco, Alana L. Oliveira, Maria de Lourdes R. S. Cunha, Márcio G. Ribeiro

**Affiliations:** 1Department of Animal Production and Preventive Veterinary Medicine, School of Veterinary Medicine and Animal Sciences, UNESP, Botucatu 18618-681, SP, Brazil; beatriz.almeida@unesp.br (B.O.d.A.); andremota5@hotmail.com (A.R.M.); ca_lechinski@yahoo.com.br (C.L.d.P.); fernando.listoni@unesp.br (F.J.P.L.); marcio.ribeiro@unesp.br (M.G.R.); 2Department of Veterinary Clinical Sciences, School of Veterinary Medicine and Animal Sciences, UNESP, Botucatu 18618-681, SP, Brazil; juliano.nobrega@unesp.br; 3Department of Chemical and Biological Sciences, Sector of Microbiology and Immunology, Institute of Biosciences, UNESP, Botucatu 18618-689, SP, Brazilalana.lucena@unesp.br (A.L.O.); mlrs.cunha@unesp.br (M.d.L.R.S.C.)

**Keywords:** 16S rRNA gene, canine oral microbiota, dog bites, MALDI-TOF MS, *mec*A gene, large-scale sequencing

## Abstract

**Simple Summary:**

The diet and contact with different environments in the practice of raising dogs contribute to a complexity of microorganism that inhabit the oral cavity of these companion animals. In addition, the close exposure of humans to pets may favor the transmission of opportunistic pathogens from dogs to owners, particularly by bite attacks, in addition to manifestations of affection such as the habit of licking the owners. Considering this scenario, we investigated the microorganisms that inhabit the oral cavities of 100 healthy dogs using a combination of traditional microbiological culture and next-generation diagnostic methods. A complexity of bacteria and fungi/yeasts was identified in the oral cavities of dogs, including agents that have been described infecting humans, such as *Staphylococcus pseudintermedius* and *Pasteurella*, *Fusobacterium*, and *Capnocytophaga* species. Furthermore, bacterial isolates with multiple resistance to antimicrobials used in human and/or animal treatment protocols were observed as well. Overall, we identified a complex microbiota inhabiting the oral cavity of healthy dogs, highlighting the risks of transmission of microorganisms from dogs to humans, especially secondary to bites, and the challenges in therapeutic approaches for humans bitten.

**Abstract:**

The high complexity of the oral microbiota of healthy dogs and the close exposure of humans to companion animals represent a risk of the transmission of potential zoonotic microorganisms to humans, especially through dog bites, including multidrug-resistant ones. Nonetheless, a limited number of comprehensive studies have focused on the diversity of the microorganisms that inhabit the oral cavities of healthy dogs, particularly based on modern molecular techniques. We investigated bacterial and fungal organisms in the oral cavities of 100 healthy dogs based on a combination of conventional and selective microbiological culture, mass spectrometry (MALDI-TOF MS), and next-generation sequencing. In addition, in vitro antimicrobial susceptibility patterns of isolates and *mec*A resistance gene were assessed. A total of 213 bacteria and 20 fungi were isolated. *Staphylococcus pseudintermedius* (40/100 = 40%), α-hemolytic *Streptococcus* (37/100 = 37%), and *Pasteurella stomatis* (22/100 = 22%) were the most prevalent bacteria diagnosed by microbiological culture and MALDI-TOF MS, whereas *Aspergillus* (10/100 = 10%) was the most common fungi identified. Based on next-generation sequencing of selected 20 sampled dogs, *Porphyromonas* (32.5%), *Moraxella* (16.3%), *Fusobacterium* (12.8%), *Conchiformibius* (9.5%), *Bergeyella* (5%), *Campylobacter* (3.8%), and *Capnocytophaga* (3.4%) genera were prevalent. A high multidrug resistance rate was observed in *Staphylococcus pseudintermedius* isolates, particularly to azithromycin (19/19 = 100%), penicillin (15/19 = 78.9%), and sulfamethoxazole/trimethoprim (15/19 = 78.9%). In addition, the *mec*A resistance gene was detected in 6.1% (3/49) of coagulase-positive staphylococci. Here, we highlight the microbial complexity of the oral mucosa of healthy dogs, including potential zoonotic microorganisms and multidrug-resistant bacteria, contributing with the investigation of the microbiota and antimicrobial resistance patterns of the microorganisms that inhabit the oral cavity of healthy dogs.

## 1. Introduction

The population of dogs and cats has significantly increased worldwide. The close contact of these pets with their owners may favor the transmission of some microorganisms, particularly those with zoonotic potential, which is a public health issue [1].

The oral cavity of dogs is one of the body regions inhabited by a wide variety of microorganisms, including aerobic and anaerobic bacteria [1]. Despite the complex microbiota of the oral cavities of dogs, including well-known zoonotic bacteria, e.g., *Pasteurella*, *Staphylococcus*, *Capnocytophaga*, and *Fusobacterium* species, a restricted number of comprehensive studies have focused on the complexity of the aerobic, anaerobic, and fungal/yeast microbiota that resides in the oral cavities of healthy dogs [2,3].

Similarities observed between microorganisms isolated from the oral mucosa of dogs and bite wounds in humans from the USA revealed the potential risks of transmission of opportunistic pathogens from companion animals to owners [1]. In Brazil, 6707 cases of aggression by dogs were reported between 2007 and 2011, which reinforces the concern regarding the transmission of these microorganisms through dog bites [4].

Given the complexity of the microbiota of the oral cavities of dogs, the wounds caused by dog bites in humans are infected by multiple species, which makes treatment difficult [5]. Approximately 20% of human accidents caused by dog bites have been related to complications due to bacterial infections, particularly in cases with high skin lacerations [6]. Endocarditis, meningitis, brain abscess, and sepsis have been reported as complications after canine bites, mainly among immunosuppressed individuals [7].

The increase in bacterial multidrug resistance to conventional antimicrobials is an emergent issue worldwide [8]. However, only a few comprehensive studies have focused on the identification of multidrug-resistant isolates and the complexity of the microorganisms that inhabit the oral cavities of healthy dogs, particularly based on modern molecular techniques, including mass spectrometry and next-generation sequencing [1,9].

Considering this scenario, we investigated the oral microbiota of 100 healthy dogs, including potential zoonotic microorganisms, where traditional and selective microbiological culture procedures, as well as mass spectrometry, next-generation sequencing, and in vitro multidrug resistance patterns of bacterial isolates were assessed for diagnostic purposes.

## 2. Materials and Methods

### 2.1. Animal Use Ethics Committee

This study was approved by the Animal Use Ethics Committee (CEUA) guidelines of FMVZ-UNESP/Botucatu, SP, Brazil, protocol number 0077/2018.

### 2.2. Animals and Sampling

A convenience sample of 100 apparently healthy dogs in a city located in the central region of the state of São Paulo, Brazil was used.

Dogs were subjected to a triplicate collection of swabs from the oral cavity (one for stock and two for microbiological cultures) using physical restraint. The sampling of dogs was performed at the home of owners by the friction of the sterile swabs against the animals’ teeth, gums, tongue, and hard palate. Next, two swabs were stored in Stuart transport medium and another one was stored in a microtube with sterile saline solution. All the samples were kept in refrigeration condition (4–8 °C) and immediately subjected to microbiology diagnosis. The samples destined for stock were vortexed, frozen at −20 °C with glycerol (10%), and kept for two years.

The animals were eligible to study if they meet the following conditions: (i) no administration of antimicrobials, by any route, in the last 30 days (based on the date of sampling), (ii) absence of periodontal signs, and (iii) absence of any systemic and/or cutaneous disease.

### 2.3. Bacteriological and Fungal/Yeast Culture

No treatment was carried out previously on the microbiological culture of samples. All oral samples were directly cultured simultaneously under aerobic and microaerophilic (5% CO_2_) conditions in sheep blood agar (5%) (Oxoid^®^, São Paulo, Brazil) [10]. The plates were incubated at 37 °C for 72 h. In addition, the samples were subjected to aerobic culture in selective MacConkey agar medium (Oxoid^®^, São Paulo, Brazil) [11] under the same conditions described above, and in Sabouraud agar (Oxoid^®^, São Paulo, Brazil) [12] with the addition of chloramphenicol (25 mg/500 mL) and incubated at 37 °C for 14 days. The bacterial and fungal/yeast microorganisms were classified based on their conventional macro- and micro-phenotypic characteristics (i.e., the morphology of colonies, Gram, and methylene blue staining) [13] and biochemical tests for enterobacteria species [14]. The coagulase-positive staphylococci isolates were subjected to further diagnosis by mass spectrometry. At least 5 colony-forming units (CFUs) of similar colonies of the same bacterial genus were considered for diagnostic purposes.

### 2.4. Selective Isolation of Pasteurella spp.

All samples were cultured on sheep blood agar medium containing 6 µg of vancomycin for selective isolation of *Pasteurella* sp. [15].

### 2.5. Selective Isolation of Staphylococcus aureus and Identification of Methicillin-Resistant Isolates (Expressing the mecA Gene)

All oral samples were cultured in egg-tellurite-glycine-pyruvate agar (BD Baird-Parker Agar^®^, Heidelberg, Germany) with the addition of egg yolk with potassium tellurite (EY Tellurite Enrichment, Difco^TM^, Heidelberg, Germany) and incubated in aerobic conditions at 37 °C for 48 h for selective isolation of staphylococci.

Dark colonies with double lipolysis halos were submitted to conventional biochemichal tests (i.e., coagulase, mannitol salt agar, DNase and trehalose) for classification and further subjected to mass spectrometry for species confirmation.

Isolates identified as *Staphylococcus aureus*, *S. intermedius*, and *S. pseudintermedius* by mass spectrometry were submitted to the detection of the *mec*A gene according to Murakami et al. (1991) [16]. Two to three typical colonies of each isolate were suspended in 10 mM Tris-HCl-1 mM EDTA (pH 8.0) at a density of ~3 × 10^8^ CFU/mL. Ten microliters of achromopeptidase (10,000 U/mL; Wako Pure Chemical Industries, Ltd., Osaka, Japan) were added to 240 µL of the bacterial suspension and incubated at 55 °C for 30 min. Then, 250 µL of the buffer and 2.5 µL of the 20% sodium dodecyl sulfate were added to lyse bacterial isolates. After 10 min of incubation at 100 °C, the lysate was centrifuged at 9500× *g* for 5 min, and 5 µL of supernatant containing bacterial DNA was used for PCR. The PCR reactions were performed in 0.2 mL microcentrifuge tubes with a total volume of 25 µL containing 10 pmol of each primer (*mec*A1 = 5′ AAA ATC GAT GGT AAA GGT TGGC, 533 pb, and *mec*A2 = 5′ AGT TCT GCA GTA CCG GAT TTGC), 2.5 U Taq DNA polymerase, 200 µL deoxyribonucleotide triphosphates, 20 mM Tris-HCl (pH 8.4), 0.75 mM MgCl_2_, and 3 µL MasterCyclerTM DNA gradient (Eppendorf, Hamburg, Germany) using the following conditions: 40 cycles of denaturation at 94 °C for 30 s, annealing of the primers at 55 °C for 30 s, and extension at 72 °C for one minute. For every reaction, international reference strains were used as positive (*S. aureus* ATCC 33591) and negative (*S. aureus* ATCC 25923) controls. The efficiency of the amplification was monitored by electrophoresis in a 2% agarose gel and stained with Syber Safe^TM^ (Invitrogen, Carlsbad, CA, USA). The amplified DNA fragments were visualized and photographed in an ultraviolet transilluminator.

### 2.6. Matrix-Assisted Laser Desorption Ionization Time-of-Flight Mass Spectrometry—MALDI-TOF MS

All isolates compatible with *Pasteurella* sp. and coagulase-positive *Staphylococcus* sp. were subjected to MALDI-TOF MS analysis in a Bruker Autoflex (Bruker Daltonik^TM^, Bremen, Germany) under 337 nm nitrogen laser conditions using FlexControl 3.3 software (Bruker Daltonik^TM^, Bremen, Germany). Spectra were analyzed between 2000 and 20,000 *m*/*z* using MALDI Biotyper 2.0 software (Bruker Daltonik^TM^, Bremen, Germany) at default settings, and identification at the genus and species levels was considered ≥1.7 and ≥2.0, respectively [17].

### 2.7. In Vitro Antimicrobial Susceptibility Profile

Antimicrobial susceptibility testing of bacterial isolates was performed using the disk diffusion method with bacterial suspensions equivalent to a 0.5 McFarland standard, according to the recommendations of the Clinical Laboratory Standards Institute [18,19]. Twelve antibiotic disks (Oxoid, Basingstoke, UK) belonging to six classes of drugs were used as follows: (1) aminoglycosides (gentamicin, 10 µg); (2) amphenicols (chloramphenicol, 30 µg); (3) fluoroquinolones (enrofloxacin, 5 µg; ciprofloxacin, 5 µg; levofloxacin, 5 µg); (4) macrolides (azithromycin, 15 µg); (5) penicillin and beta-lactam derivatives (penicillin, 10 IU; ceftriaxone, 30 µg; ceftiofur, 30 µg; amoxicillin/clavulanic acid, 30 µg; imipenem, 10 µg); and (6) potentiated sulfonamides (sulfamethoxazole/trimethoprim, 25 µg). The multidrug resistance profile was classified into seven groups of bacterial isolates, as follows: *Staphylococcus* spp., *Pasteurella* spp., α-hemolytic *Streptococcus*, *Neisseria* spp., enterobacteria, *Pseudomonas* sp., and miscellaneous, where susceptibility of isolates was based on antimicrobials indicated for each group [18,19] considering intrinsic resistance of some bacteria. Isolates exhibiting resistance to ≥3 different groups of antimicrobials indicated to each group were considered multiresistant [20]. Intermediate results of in vitro antimicrobial pattern were not included in analysis of multiresistant isolates.

### 2.8. Next-Generation Sequencing

Next-generation sequencing of the oral microbiota was performed in 20 dogs randomly selected (one out of five dogs sampled in the study). The samples were collected using appropriate swabs and microtubes provided by the private laboratory Neoprospecta, Santa Catarina, Brazil. The V3-V4 regions of the 16S rRNA gene were amplified using the primers 341F (CCTACGGGRSGCAGCAG) and 806R (GGACTACHVGGGTWTCTAAT).

All the PCR reactions were carried out in triplicates using Platinum Taq (Invitrogen, USA) with the following conditions: 95 °C for 5 min, 25 cycles of 95 °C for 45 s, 55 °C for 30 s, 72 °C for 45 s, and a final extension of 72 °C for 2 min for the first PCR. For the second PCR, the conditions were 95 °C for 5 min, 10 cycles of 95 °C for 45 s, 66 °C for 30 s, and 72 °C for 45 s, and a final extension of 72 °C for 2 min. For comparison, the Illumina 16S protocol was used. The final PCR reaction was cleaned up using AMPureXP beads (Beckman Coulter, Brea, CA, USA), and samples were pooled in the sequencing libraries for quantification. The pool amplicon estimations were performed with Picogreen dsDNA assays (Invitrogen, USA), and then the pooled libraries were diluted for accurate qPCR quantification using KAPA Library Quantification Kit for Illumina platforms (KAPA Biosystems, Woburn, MA, USA).

The libraries were sequenced in a MiSeq Sequencing System, using the standard Illumina primers provided in the kit. Usually, a single-end 300 nt run was performed.

## 3. Results

### 3.1. Microorganism Identification

A total of 213 bacterial isolates were obtained from the oral cavity samples. *Staphylococcus pseudintermedius* (40/213 = 18.78%), α-hemolytic *Streptococcus* (37/213 = 17.37%), *Pasteurella stomatis* (22/213 = 10.33%), coagulase-negative *Staphylococcus* spp. (18/213 = 8.45%), *Pasteurella canis* (16/213 = 7.51%), *Pseudomonas* sp. (12/213 = 5.63%), *Escherichia coli* (11/213 = 5.16%), *Neisseria animaloris* (10/213 = 4.69%), and *Neisseria zoodegmatis* (10/231 = 4.69%) were the main microorganisms identified by microbiological culture and/or MALDI-TOF MS (Table 1).

#### 3.1.1. *Pasteurella* and *Neisseria* Species

Colonies compatible with *Pasteurella* spp. were isolated from 60% (60/100) of the oral cavity samples. Two different types of *Pasteurella* colonies were isolated in four oral samples. A total of 64 isolates of *Pasteurella* spp. were identified. MALDI-TOF MS revealed that *P. stomatis* was the most frequent species (22/64 = 34.4%), followed by *P. canis* (16/64 = 25%), *P. multocida* (3/64 = 4.7%), and *P. dagmatis* (2/64 = 3.1%). The remaining isolates were identified as *Neisseria* species (21/64 = 32.8%), including *N. animaloris* (10/64 = 15.6%), *N. zoodegmatis* (10/64 = 15.6%), and *N. weaveri* (1/64 = 1.6%). The prevalence of the genus *Pasteurella* was 42% (42/100), since two distinct species were identified in a sample from the same oral cavity, whereas the prevalence of the genus *Neisseria* was 19% (19/100), as two samples revealed two different species.

#### 3.1.2. Staphylococci

Typical colonies of staphylococci were isolated in 66% (66/100) of the animals sampled. One oral sample revealed two different types of colonies, totaling 67 isolates. Dark colonies with double lipolysis halos were preliminarily classified as *S. aureus* after conventional biochemical tests (i.e., coagulase, mannitol salt agar, DNase, and trehalose). Coagulase-positive and coagulase-negative isolates were identified in 73.1% (49/67) and 26.9% (18/67) of staphylococci isolates, respectively. MALDI-TOF MS revealed that most of the coagulase-positive isolates, previously classified as *S. aureus*, were identified as *S. pseudintermedius* (40/49 = 81.6%), followed by *S. intermedius* (6/49 = 12.3%) and *S. aureus* (3/49 = 6.1%), whereas coagulase-negative isolates were classified as *Staphylococcus* spp.

#### 3.1.3. Enterobacteria

Enterobacteria species were isolated among 21% (21/100) of dogs sampled. Two different colonies of enterobacteria were isolated from two oral samples, totaling 23 isolates. *Escherichia coli* (11/23 = 47.8%), *Enterobacter cloacae* (4/23 = 17.4%), *Klebsiella pneumoniae* (3/23 = 13%), *Enterobacter aerogenes* (2/23 = 8.7%), *Proteus mirabilis* (2/23 = 8.7%), and *Citrobacter freundii* (1/23 = 4.4%) were the identified species.

#### 3.1.4. Miscellaneous

Other bacterial microorganisms isolated from oral cavities of dogs were alpha-hemolytic *Streptococcus* (37/100 = 37%), *Pseudomonas* sp. (12/100 = 12%), and *Chryseobacterium* spp. (2/100 = 2%).

#### 3.1.5. Fungi and Yeasts

Fungi and yeasts were isolated in 19% of oral samples (19/100). One sample exhibited two distinct colonies, resulting in 20 isolates. *Aspergillus* (10/20 = 50%) was the most prevalent genus, followed by *Malassezia pachydermatis* (4/20 = 20%). Less frequently identified were the genera *Bipolaris* (1/20 = 5%), *Curvularia* (1/20 = 5%), *Fusarium* (1/20 = 5%), *Nigrospora* (1/20 = 5%), *Scopulariopsis* (1/20 = 5%), and *Trichoderma* (1/20 = 5%).

### 3.2. In Vitro Antimicrobial Susceptibility Test

Amoxicillin/clavulanic acid was the most effective drug against the main groups of bacteria identified, except enterobacteria, in which chloramphenicol showed better efficacy. In general, *Pasteurella* and *Neisseria* isolates showed little resistance to the studied drugs, whereas staphylococci showed resistance to various antimicrobials in variable frequencies (Table A1—Appendix A).

Bacterial multidrug resistance was observed in 13.1% (28/213) of isolates, identified as *Staphylococcus pseudintermedius* (19/40 = 47.5%), *Staphylococcus* spp. (4/18 = 22.2%), *Chryseobacterium* spp. (2/2 = 100%), *Escherichia coli* (2/11 = 18.2%), and *Staphylococcus aureus* (1/3 = 33.3%).

The *mec*A gene was detected in 6.1% (3/49) of coagulase-positive *Staphylococcus* isolates, i.e., one *S. aureus* and two *S. pseudintermedius* (Figure A1—Appendix A).

### 3.3. Next-Generation Sequencing

Large-scale sequencing performed in 20 oral cavity samples of healthy dogs detected a complexity of microorganism species, which varied between 3794 and 101,622 sequences in each sample, with a total of 1,071,241 and an average of 53,562 sequences/sample (BioProject PRJNA836950) (Figure A2 and Figure A3—Appendix A).

Oral samples from 20 dogs subjected to next-generation sequencing exhibited a high relative abundance of *Moraxella* spp. (174,860/1,071,241 = 16.32%), *Bergeyella* spp. (53,150/1,071,241 = 4.96%), *Campylobacter* spp. (40,909/1,071,241 = 3.81%), *Capnocytophaga* spp. (36,313/1,071,241 = 3.38%), and *Corynebacterium* spp. (22,521/1,071,241 = 2.1%). *Pasteurella* and *Neisseria* were detected among the 10 genera with highest relative abundance in all samples (25,980/1,071,241 = 2.42% and 36,219/1,071,241 = 3.38%, respectively).

*Fusobacterium* and *Porphyromonas*, two anaerobic genera of microorganisms, were detected in a high relative abundance among the 20 oral samples studied (137,102/1,071,241 = 12.79% and 348,601/1,071,241 = 32.54%, respectively).

## 4. Discussion

This study revealed a great complexity of bacterial and fungal/yeast microorganisms, including agents that have been described infecting humans, in the oral cavity of 100 apparently healthy dogs using conventional and selective microbiological culture and advanced molecular techniques (MALDI-TOF MS and next-generation sequencing), in addition to identification of multidrug-resistant bacteria.

The use of selective culture media enabled the isolation of species of interest, e.g., *Pasteurella* and *Staphylococcus* spp. MALDI-TOF MS was used for the species-level identification of isolates compatible with *Pasteurella* and *Staphylococcus* species, which allowed the detection of uncommon species reported as inhabitants of the oral microbiota of healthy dogs, i.e., *Rothia nasimurium*, *Chryseobacterium gleum*, and *C. indologenes* [1,21]. In addition, the identification of microorganisms with zoonotic potential inhabiting the oral microbiota of healthy dogs deserves attention for its relevance to humans, due to the possibility of transmission of opportunistic pathogens from dogs to humans, secondary to bite attacks favored by the close exposure of owners and their dogs [5].

Among the *Pasteurella* species, *P. multocida* and *P. canis* have been reported as frequent inhabitants of the oral microbiota of dogs [22]. In the current study, mass spectrometry identified a higher prevalence of *P. stomatis* in dogs sampled, which has not been frequently observed. Despite the low prevalence, the identification of *P. multocida* in the oral cavity of healthy dogs represents a human concern, since this species has already been reported as a primary cause of some complications in people who were bitten by dogs or lived in the same environment as them. In this regard, a case of meningitis caused by *P. multocida* was reported in a child in England after contact with the hand of another person previously licked by a dog [23].

*Neisseria* species, such as *N. animaloris* and *N. zoodegmatis,* have also been reported as common inhabitants of the oral cavity of dogs. Nonetheless, these microorganisms have been described as the primary cause of pulmonary signs in dogs, and clinical complications in different organs in humans. A retrospective study in Sweden identified *N. animaloris* and *N. zoodegmatis* using microbiological culture in samples previously obtained from wounds in 13 human patients bitten by dogs, including one that evolved to septicemia [24]. Interestingly, *N. animaloris* and *N. zoodegmatis* were the most prevalent species of *Neisseria* identified herein by mass spectrometry, indicating a zoonotic potential of these opportunistic agents that naturally inhabit the oral cavities of healthy dogs.

*Staphylococcus pseudintermedius,* belonging to the *Staphylococcus intermedius* group (SIG), is a well-known inhabitant of the skin and mucosa of dogs and humans [25]. However, due to biochemical similarity with other staphylococci species, *S. pseudintermedius* has probably been underdiagnosed or underestimated as a primary agent of infection in humans bitten by dogs [26]. Four cases of infected wounds of humans from Spain caused by *S. pseudintermedius* revealed that two patients had isolates with identical patterns in electrophoretic profile, and a similar in vitro antimicrobial susceptibility test compared to the isolates obtained from their domestic dogs [27], highlighting the potential zoonotic nature of these opportunistic pathogens.

The same misdiagnosis of coagulase-positive staphylococci species was reported by Chuang et al. (2010) in China [28], in a case of septicemia-related *S. pseudintermedius* in a child with history of continuous use of an intravenous catheter for therapeutic purposes and recent contact with two dogs, reinforcing the use of molecular techniques to distinguish these species.

The low prevalence of enterobacteria isolated from the oral cavities among the studied dogs was expected because of the fecal origin of this group of bacteria. However, the isolation of *Escherichia coli* and *Klebsiella pneumoniae* in dogs sampled may be attributed to the habit of hygiene of the perineal region, as well as cleaning the offspring with the tongue in the lactating females or coprophagy in young dogs, which could facilitate the transient presence of enterobacteria in the oral cavity. In Spain, an elderly woman died after developing fulminant hemorrhagic purpura secondary to a dog bite, with extensive areas of cutaneous necrosis, where *E. coli* and *Staphylococcus warneri* coinfection was diagnosed in blood culture [29]. No report of human septicemia by *K. pneumoniae* secondary to a dog bite has been described, although it is known that this bacterium commonly shows multidrug resistance to conventional antimicrobials [30]. Therefore, the identification of *E. coli* and *K. pneumoniae* in the oral mucosa of healthy dogs in the current study reinforces the risk of enterobacteria-induced infections in humans secondary to dog bites.

α-hemolytic *Streptococcus* was identified in approximately 17% of the oral cavities of the healthy dogs sampled. However, a retrospective study performed between 1995 and 1996 in 18 emergency departments in the United States reported that 46% (23/50) of wounds in humans bitten by dogs had been caused by hemolytic streptococci, confirming the zoonotic nature of this group of microorganisms [31].

Despite the low prevalence, the identification of *Pseudomonas* sp. in the oral microbiota of healthy dogs reinforces the need for an appropriate therapeutic approach to bites caused by dogs, since *P. aeruginosa* has been related to severe nosocomial human infections, commonly refractory to conventional antimicrobial therapy and consequent poor prognosis [32].

*Chryseobacterium gleum* and *C. indologenes*, formerly belonging to the *Flavobacterium* genus, are multidrug-resistant bacteria often described in severe nosocomial infections in humans, particularly immunosuppressed patients and/or patients with a history of use of catheters and tracheotomy tubes. In addition, these species have been considered the primary causes of emerging infections in the USA [33]. Although *C. indologenes* commonly presents low pathogenicity in immunocompetent individuals, it has been previously described as an agent of septicemia in a patient from India without a history of hospitalization, invasive procedures, or immunosuppressive conditions [21]. Despite the low prevalence of this opportunistic pathogen in our study, the identification of two isolates of *Chryseobacterium* spp. in the oral cavities of healthy dogs indicates a potential risk of transmission of these microorganisms from dogs to humans.

Based on next-generation sequencing, *Neisseria* genus has been described in the oral cavities of four dogs and their owners in South Korea [34]. *Bergeyella* spp., which are inhabitant bacteria of the upper respiratory tract and the oral cavity of dogs [35], have also been identified as the primary cause of septicemia in a patient bitten by a dog in Spain [36], highlighting the concern of the cutaneous injuries in humans caused by this opportunistic pathogen.

The first report of *Capnocytophaga canimorsus* in the saliva of dogs occurred in the USA in 1978 [37] and, subsequently, other cases of serious infections were described in human patients after dog bites. A retrospective study of 484 cases of human infections by *C. canimorsus* between 1990 and 2014 revealed a history of dog bites in 60% of cases and dog scratches or licks in 27% of them, with a mortality rate of 26% [38]. In the current study, all the 20 oral samples of healthy dogs subjected to next-generation sequencing revealed the identification of *Capnocytophaga* spp. Therefore, *C. canimorsus*-induced infections in humans after a dog bite, scratch, or lick represent a human concern because of the high frequency and severity of clinical signs.

*Moraxella* sp., *Campylobacter* sp., and *Corynebacterium* sp. identified in dogs sampled have also been previously identified by amplicon sequencing in the oral cavities of dogs [39] and in human wound secretions secondary to canine bites [31,40], reinforcing the complexity of opportunistic pathogens that can be harbored by the oral cavity of these companion animals.

Besides the identification of anaerobes *Fusobacterium* and *Porphyromonas* organisms using next-generation sequencing, there was no identification of *Clostridium* species among the studied dogs. Conversely, it was identified in the USA in studies using microbiological culture from human wounds caused by dog bites [41] and next-generation sequencing in dog oral cavities [3], indicating that anaerobic agents should be considered in the diagnosis of skin lesions in humans after dog bites and should be focused on in similar further studies.

Amoxicillin/clavulanic acid was the most efficient antibiotic in our study and has been indicated for the treatment of oral infections in dogs and dog bite wounds in humans [1,2]. In fact, when it is not possible to carry out in vitro susceptibility pattern tests, this drug has been considered an option to treat human patients who suffered bite attacks, particularly by homeless dogs or those that evaded capture after attacking. Despite the high in vitro susceptibility pattern of isolates, it is important to consider a combination of antimicrobials due to the large number of different microorganisms from the mouths of dogs that may be transmitted to humans [2].

Bacterial resistance is an emerging issue on a global level [42] and has been attributed, in part, to continuous or non-judicious use of drugs in veterinary practice [43]. The identification of multidrug-resistant isolates in the oral cavities of sampled dogs, particularly *Staphylococcus pseudintermedius*, highlights the risk of transmission of resistant bacteria from the oral microbiota of dogs to humans, a concern also reported by Bata et al. (2020) [10]. In addition, this finding reinforces that, if possible, the treatment of dog bite wounds in humans should be supported by previous etiological identification and in vitro antimicrobial susceptibility patterns of isolates.

The methicillin-resistance gene (*mec*A) is a gene that encodes a penicillin-binding protein (PBP) with low affinity for oxacillin and other antimicrobials from the β-lactam group and is considered one of the main drug resistance mechanisms of the staphylococci group [44]. The refractiveness to therapy of patients infected with methicillin-resistant *Staphylococcus* (MRS) determines long hospitalization periods and high mortality outcomes [45]. The *mec*A resistance gene has already been described in *S. aureus* from dogs around the world [46]. Fessler et al. (2018) [47] also reported methicillin-resistant *S. aureus* and *S. pseudintermedius* (MRSA and MRSP, respectively) isolates in veterinary hospital employees in Germany. These findings highlight the risk of dogs harboring resistant staphylococci species that could potentially be transmitted from dogs to humans in veterinary medicine practices or secondary to nosocomial infections.

*Aspergillus* sp. are well-known saprophytic fungal organisms commonly involved in otitis and sinusitis cases in dogs, which can evolve to severe central nervous system complications [48]. Here, 10% of dogs had *Aspergillus* sp. in the oral microbiota. However, to date, no isolation of *Aspergillus* sp. from the oral cavity of healthy dogs has been reported, which makes it difficult to compare our findings. In addition, *Malassezia pachydermatis* was the only species of yeast identified in 4% of the oral cavities of dogs sampled, as opposed to other studies in Brazil, in which this agent was described in up to 30% of oral samples of dogs [49].

The discrepancies of microorganisms identified in different studies from oral microbiota of dogs, including fungi, yeasts, and anaerobic organisms, could be influenced by differences in diet and environmental conditions of raising pets [50,51], which unfortunately were not assessed in our sampled dogs.

A convenience sample of dogs, the absence of specific culture media to anaerobes, the next-generation sequencing techniques applied to only 20% of dogs sampled, and no species-level identification by MALDI-TOF MS or any other molecular system diagnosis method of fungal isolates may be considered limitations of the current study.

## 5. Conclusions

Overall, a complex microbiota was identified in the oral mucosa of dogs sampled, including multidrug-resistant bacteria and organisms with zoonotic potential, contributing with the investigation of the microbiota and antimicrobial resistance patterns of the microorganisms that inhabit the oral cavities of healthy dogs.

## Figures and Tables

**Table 1 animals-13-02467-t001:** Frequency of bacteria identified from the oral cavity of 100 healthy dogs, diagnosed by microbiological ^1^ and MALDI-TOF MS ^2^ approaches.

Microorganisms	N	%
*Staphylococcus pseudintermedius* ^2^	40	18.78%
α-hemolytic *Streptococcus* ^1^	37	17.37%
*Pasteurella stomatis* ^2^	22	10.33%
*Staphylococcus* spp. ^1^	18	8.45%
*Pasteurella canis* ^2^	16	7.51%
*Pseudomonas* sp. ^1^	12	5.63%
*Escherichia coli* ^1^	11	5.16%
*Neisseria animaloris* ^2^	10	4.69%
*Neisseria zoodegmatis* ^2^	10	4.69%
*Staphylococcus intermedius* ^2^	6	2.82%
*Enterobacter cloacae* ^1^	4	1.88%
*Micrococcus* spp. ^1^	4	1.88%
*Rothia nasimurium* ^2^	4	1.88%
*Klebsiella pneumoniae* ^1^	3	1.41%
*Pasteurella multocida* ^2^	3	1.41%
*Staphylococcus aureus* ^2^	3	1.41%
*Enterobacter aerogenes* ^1^	2	0.94%
*Pasteurella dagmatis* ^2^	2	0.94%
*Proteus mirabilis* ^1^	2	0.94%
*Chryseobacterium gleum* ^2^	1	0.47%
*Chryseobacterium indologenes* ^2^	1	0.47%
*Citrobacter freundii* ^1^	1	0.47%
*Neisseria weaveri* ^2^	1	0.47%
Total	213	100.00%

MALDI-TOF MS = Matrix-Assisted Laser Desorption Ionization—Time of flight mass spectrometry; N = Number of isolates; % = Frequency of isolates.

## Data Availability

Next-generation sequencing data of the 20 sampled dogs can be found in National Center for Biotechnology Information—NCBI BioProject PRJNA836950 (https://www.ncbi.nlm.nih.gov/bioproject/?term=PRJNA836950 (accessed on 10 May 2022).

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
