# Peer review of "Microbial Complexity of Oral Cavity of Healthy Dogs Identified by Mass Spectrometry and Next-Generation Sequencing"

_animals, 2023, doi:10.3390/ani13152467_

Round 1

Reviewer 1 Report

1- Why the author did not yield strict anaerobic strains during their isolation? This fact suggested that the conservation of the samples prior the isolation of anaerobes is not well performed. 

Generally anarobes requires some kind of nutricional supplements  not only blood to grow. It may be better to use especific anaerobes media.

Next generation sequencing found strict anarobes, so anaerobes may be detected if apropiated media were used.

Authors do not discus this discrepancy.

Another important point, is that the study lacks the analisis of resistant pattern of anaerobic strains ( Authors ddid not isolated anarobes) and this analysis may be important.

Reviewer 2 Report

Title:

The article is very focused on the relationship between dog bite and human health risk. The title does not reveal this implicit objective of the article.

Simple Summary

The diet habits and contact with different environments in breeding practice 

What do you mean by breeding practice in this context?

Line 17 - But also through manifestations of affection such as the habit of licking the owners.

Introduction

Line 81- with zoonotic behavior,...What do you mean by "zoonotic behavior?

Results

Line 191 to 198 - Scientific names italicized

Line 292 - statistical p in italics

Discussion

Line 344 - Animals also have hygiene habits that involve the perineal region and are therefore contaminated with feces, just as lactating females clean their offspring with their tongue.

Line 364 - nosocomial instead hospital-acquired

Reference suggestion 

Clinical and mycological analysis of dog's oral cavity.Santin R, Mattei AS, Waller SB, Madrid IM, Cleff MB, Xavier MO, de Oliveira Nobre M, Nascente Pda S, de Mello JR, Meireles MC.Braz J Microbiol. 2013 Apr 9;44(1):139-43. doi: 10.1590/S1517-83822013005000018. eCollection 2013.

Cultivable oral microbiota of domestic dogs. Elliott DR, Wilson M, Buckley CM, Spratt DA. J Clin Microbiol. 2005 Nov;43(11):5470-6. doi: 10.1128/JCM.43.11.5470-5476.2005

Gene Sequence Analyses of the Healthy Oral Microbiome in Humans and Companion Animals. Davis EM. J Vet Dent. 2016 Jun;33(2):97-107. doi: 10.1177/0898756416657239. Epub 2016 Aug 6.

Intra-oral microbial profiles of beagle dogs assessed by checkerboard DNA-DNA hybridization using human probes.Rober M, Quirynen M, Haffajee AD, Schepers E, Teughels W.Vet Microbiol. 2008 Feb 5;127(1-2):79-88. doi: 10.1016/j.vetmic.2007.08.007. Epub 2007 Aug 15.

Characterising the Canine Oral Microbiome by Direct Sequencing of Reverse-Transcribed rRNA Molecules. McDonald JE, Larsen N, Pennington A, Connolly J, Wallis C, Rooks DJ, Hall N, McCarthy AJ, Allison HE.PLoS One. 2016 Jun 8;11(6):e0157046. doi: 10.1371/journal.pone.0157046. eCollection 2016.

Gingival flora of the dog with special reference to bacteria associated with bites. Saphir DA, Carter GR. J Clin Microbiol. 1976 Mar;3(3):344-9. doi: 10.1128/jcm.3.3.344-349.1976.

Metagenomic analysis of the canine oral cavity as revealed by high-throughput pyrosequencing of the 16S rRNA gene. Sturgeon A, Stull JW, Costa MC, Weese JS. Vet Microbiol. 2013 Mar 23;162(2-4):891-898. doi: 10.1016/j.vetmic.2012.11.018. Epub 2012 Nov 20.

Study of microbiocenosis of canine dental biofilms. Kačírová J, Maďari A, Mucha R, Fecskeová LK, Mujakic I, Koblížek M, Nemcová R, Maďar M.Sci Rep. 2021 Oct 5;11(1):19776. doi: 10.1038/s41598-021-99342-5.
